# UV-B Radiation Exhibited Tissue-Specific Regulation of Isoflavone Biosynthesis in Soybean Cell Suspension Cultures

**DOI:** 10.3390/foods13152385

**Published:** 2024-07-28

**Authors:** Mian Wang, Yiting Wang, Muhammad Bilal, Chong Xie, Pei Wang, Xin Rui, Runqiang Yang

**Affiliations:** 1College of Food Science and Technology, Whole Grain Food Engineering Research Center, Nanjing Agricultural University, Nanjing 210095, China; 2018108018@njau.edu.cn (M.W.); 2023908007@stu.njau.edu.cn (Y.W.); 2022108181@stu.njau.edu.cn (M.B.); xiechong@njau.edu.cn (C.X.); wangpei@njau.edu.cn (P.W.); ruix@njau.edu.cn (X.R.); 2Sanya Institute of Nanjing Agricultural University, Sanya 572024, China

**Keywords:** UV-B radiation, soybean suspension cultures, isoflavones, tissue specificity

## Abstract

Isoflavones, a class of substances with high biological activity, are abundant in soybeans. This study investigated isoflavone biosynthesis in soybean cell suspension cultures under UV-B radiation. UV-B radiation enhanced the transcription level and activity of key enzymes involved in isoflavone synthesis in cell suspension cultures. As a result, the isoflavone contents significantly increased by 19.80% and 91.21% in hypocotyl and cotyledon suspension cultures compared with the control, respectively. Meanwhile, a significant difference was observed in the composition of isoflavones between soybean hypocotyl and cotyledon suspension cultures. Genistin was only detected in hypocotyl suspension cultures, whereas glycitin, daidzein, and genistein accumulated in cotyledon suspension cultures. Therefore, UV-B radiation exhibited tissue-specific regulation of isoflavone biosynthesis in soybean cell suspension cultures. The combination of suspension cultures and abiotic stress provides a novel technological approach to isoflavone accumulation.

## 1. Introduction

In recent years, as the economy has developed, there has been a rapid improvement in people’s living standards. Consequently, there has been a growing emphasis on food nutrition and health. Functional foods have emerged as a vital part of people’s daily diet and healthy lifestyle. These foods offer various benefits, including boosting immunity, improving physiological functions, preventing chronic diseases, delaying aging, and addressing subhealth issues [1]. Also, the demand for functional foods has significantly increased due to the aging population. Therefore, the development of functional foods with beneficial components has become increasingly popular in the food industry.

Isoflavones, the main secondary metabolites in leguminous plants [2], have a protective effect against ultraviolet (UV) radiation. Isoflavones have various physiological effects on the human body, including estrogen regulation, osteoporosis prevention, anti-cancer, blood lipid reduction, and cardiovascular protection [3]. Light stress is an effective pathway for the accumulation of isoflavones, especially UV-B radiation [4]. The total isoflavone content in germinated soybeans reached approximately 8000 µg/g under UV-B stress [5]. Lim et al. [6] reported that short-term light exposure increased the levels of light-sensitive compounds in soybean seedlings, such as malonyl daidzin and malonyl genistin. Yin et al. [7] also found that UV-B radiation induced isoflavone synthesis in 4-day-old germinated soybeans, such as daidzein and genistein. Therefore, the accumulation of isoflavones through UV-B radiation enhances the nutritional value and utilization of soybeans and other grain crops. Nonetheless, the impact of UV-B radiation on soybean suspension cultures remains unclear, necessitating further investigation into whether UV-B radiation could improve the enrichment of isoflavones in suspension cells.

Traditional methods of enriching functional compounds in plants, such as germination or crop breeding [8], are often constrained by high costs and natural conditions, making large-scale production challenging. In contrast to whole plants, plant cell suspension cultures demonstrate elevated cellular metabolic rates and offer a range of benefits, including cost-effectiveness, high yields, resource conservation, reduced land requirements, and environmental sustainability [9]. It provides a promising platform for producing functional compounds and has already shown successful applications in crops like soybeans and wheat [10]. Rani and Vimolmangkang [11] reported that the total isoflavone content reached up to 46.3 mg/g DW in soybean suspension cells. Bioreactors have facilitated the sustained production of daidzein and genistein, with concentrations ranging from 10 to 200 µg/g DW, for a period exceeding two months. Jeong et al. [12] observed a high production of glycosidic isoflavones in soybean suspension cell cultures under methyl jasmonate treatment, resulting in a 10-fold increase in total isoflavones. Devi et al. [13] also investigated the biosynthesis of isoflavones in cell cultures of soybeans, and pointed out that IFS2 was the key gene responsible for isoflavone accumulation. Furthermore, the heterologous expression of the AtCPK1-Ca gene through gene editing technology increased the total isoflavone content to 208.09 mg/L in cell cultures, along with an increase in isoflavone aglycones [14]. Therefore, soybean suspension cultures hold great potential for the production of bioactive components, such as isoflavones. The isoflavone content varies across different parts of the soybean plant. The hypocotyl and cotyledon of soybeans are known to have relatively high isoflavone levels [11]. Previous studies have analyzed the composition and quantities of isoflavone monomers in soybean hypocotyl and cotyledon calluses [15]. Thus, there may be variations in the nutritional composition and metabolic levels of soybean suspension cultures induced from different explants. The accumulation of isoflavones and their monomer composition still need further analysis.

The aim of this study was to observe the response of soybean cell suspension cultures to UV-B radiation, reveal the differences in isoflavone synthesis between soybean hypocotyl and cotyledon suspension cultures, and explore the effects of UV-B radiation on the regulation of isoflavone synthesis in cell suspension cultures. These findings could present a novel biosynthesis approach for the accumulation of functional compounds, such as isoflavones. They are beneficial for reducing resource wastage and achieving a continuous production of secondary metabolites.

## 2. Materials and Methods

### 2.1. Chemicals

*Glycine max* L. seeds (Dongnong, 2021) were collected from the Jiangsu Academy of Agricultural Sciences, China. All reagents used were obtained from Aladdin (Aladdin Reagent Co., Ltd., Shanghai, China), including isoflavone standards (daidzin, glycitin, genistin, malonyl daidzin, malonyl glycitin, malonyl genistin, daidzein, glycitein, genistein (purity ≥ 90%)), sucrose, agar, Murashige and Skoog (MS) basal medium (without sugar and agar), 1/2 MS medium, methanol (HPLC grade), ethyl acetate, acetonitrile (HPLC grade), glacial acetic acid (HPLC grade), rutin, gallic acid, perchloric acid, 1,1-diphenyl-2-picrylhydrazyl, 2, 2′-azino-bis(3-ethylbenzothiazoline-6-sulfonic acid), Trolox, β-mercaptoethanol, etc.

### 2.2. Preparation of Cell Suspension Cultures

The establishment of explants and the induction of callus were performed as previously described by us [15]. The cell suspension cultures were initiated from 14-day-old soybean hypocotyl and cotyledon callus. The callus (approximately 1.0 g) was dissected into small pieces using a scalpel and propagated in MS liquid medium (containing 1.0 mg/L 2,4,-D, 0.5 mg/L 6-BA and 3% sucrose) in the dark at 25 °C by shake culturing. The cultures were passaged weekly into a fresh liquid medium at a 1:4 volume ratio. Cell suspension cultures were consistently established after 3–5 passages.

The cultures exposed to 40 μW/cm^2^ UV-B for 2 h, followed by a 12 h incubation in the dark, were the UV-B group (UV-B). Cell suspension cultures without UV-B radiation served as the control group (control). Subsequently, some cultures were freeze-dried and stored at −20 °C for further analysis. Others were stored at −80 °C until analysis.

### 2.3. Microscopic Observations

To assess the viability of cells, the soybean cell cultures were incubated with 0.04 mg/mL 3,6-diacetoxyfluoran for 30 min. Following three washes with PBS (pH 5.8, 0.01 M), cells were resuspended in a fresh culture medium. The fluorescence of viable cells was observed using Zeiss Axio imaging equipment (Borex Technology Co., Ltd., Beijing, China).

### 2.4. Cell Growth and Viability

Cell growth: The growth of suspended cells was quantified based on dry weight (DW). Cells were collected every 3 days, filtered, and recovered. The fresh cell mass was dried to a constant weight at 45 °C. The cell biomass represented the dry weight of cells harvested per 40 mL culture (g/flask).

Cell viability: The TTC method [16] was used to measure cell viability. The suspension cultures were incubated with 0.5% triphenyltetrazolium chloride for 13–16 h, then extracted with 95% ethanol until colorless. The cell viability was determined at 485 nm.

### 2.5. Physiological Metabolism

The oxidative damage level of suspension cultures was determined as described by Wang et al. [15], including MDA content and electrolyte leakage. ATP, ADP, and AMP contents were measured as described by Jin et al. [17].

The key enzyme activity of energy metabolism was determined using a succinate dehydrogenase (SDH) kit (BC0950, Solarbio Technology Co., Ltd., Beijing, China), a cytochrome oxidase (CCO) ELISA kit (SJ01680, Jining Industrial Co., Ltd., Shanghai, China), an H^+^-ATPase kit (A070-4, Jiancheng Bioengineering Institute, Nanjing, China) and a Ca^2+^-ATPase kit (A069-1, Jiancheng Bioengineering Institute, Nanjing, China), respectively. Cell suspension cultures were thoroughly ground in 9% physiological saline. The enzyme extraction solution was the supernatant obtained after centrifugation (4 °C, 3500× *g* for 15 min) for further analysis.

### 2.6. Phenolics and Flavonoid Contents and Their Antioxidant Capacity

The extraction and quantification of phenolics and flavonoids were performed as described by our previous study [18]. The antioxidant capacity of the extracts was also assessed. The contents of phenolics and flavonoids were calculated using gallic acid and rutin as equivalents, respectively. Trolox was used for the quantitative analysis of the antioxidant capacity.

### 2.7. Identification and Quantification of Isoflavonoids

The extraction of isoflavonoids was performed as previously described by us [19]. The amount of 20 mg of freeze-dried cell culture was ultrasonically extracted with 80% (*v*/*v*) methanol. The extract was then filtered through a 0.45 μm membrane.

Chromatographic separation was performed using a reversed-phase column (Ultimate AQ-C18, 4.6 × 250 mm, 5 μm particle size). The mobile phase consisted of solvent A, 0.1% acetic acid in water, and solvent B, 0.1% acetic acid in acetonitrile. A 52 min gradient was programmed as follows: 0–50 min, 13–35% B; 50–51 min, 35–13% B; 51–52 min, 13% B. The column temperature was maintained at 35 °C throughout the analysis. A sample injection volume of 20 μL was used, at a constant flow rate of 1.0 mL/min. The analytes were detected by monitoring the absorbance at 260 nm. Typical HPLC chromatograms of isoflavone monomers are shown in Appendix A.

### 2.8. Gene Expression and Enzyme Activity in Isoflavone Synthesis

The relative expression and activity of key enzymes were determined as described by us [20], including phenylalanine ammonialyase (PAL), cinnamate-4-hydroxylase (C4H), 4-coumarate-CoA ligase (4CL), chalcone ketone synthase (CHS), chalcone reductase (CHR), chalcone isomerase (CHI), and isoflavone synthase (IFS). The sequence-specific primers are listed in Table 1.

### 2.9. Statistical Analysis

Statistical analyses were performed using SPSS 18.0 (SPSS Inc., Chicago, IL, USA). All tests were three biological replicates and analyzed by Duncan’s multiple-range tests at *p*-values < 0.05.

## 3. Results

### 3.1. Physiological Indicators of Cell Suspension Cultures

Cell suspension cultures consisted of numerous cell clusters and single cells, with high fluorescence intensity and strong cellular vitality. As shown in Figure 1a,b, soybean cells displayed three distinct shapes: round, oval, and elongated, with cotyledon cells being relatively smaller than hypocotyl cells. The vitality of cell suspension cultures showed a significant increase in the early stage, reaching its peak on days 6 to 8. And the vitality gradually decreased as the cultivation time extended (Figure 1d). Correspondingly, the biomass of cell cultures significantly increased in the early stages of cultivation and entered a stable phase thereafter, accompanied by signs of cellular aging and apoptosis (Figure 1c). Notably, the vitality and biomass accumulation of soybean cotyledon cells were both higher than those of hypocotyl cells.

The physiological metabolism levels of the cell suspension cultures are shown in Figure 1e,f. The MDA content in cotyledon suspension cultures significantly increased by 60.65% under UV-B radiation. The electrolyte permeability of hypocotyl and cotyledon suspension cultures significantly increased by 36.23% and 14.84%, respectively.

### 3.2. Energy Metabolism of Cell Suspension Cultures

The energy reserves in cell suspension cultures are mainly ATP and AMP (Table 2). UV-B radiation promoted the production of ADP. There was a significant accumulation of AMP in hypocotyl suspension cultures. The ATP content only significantly increased in cotyledon suspension cultures under UV-B radiation. Correspondingly, the energy charge significantly decreased in soybean hypocotyl cultures, while it increased in soybean cotyledon cultures.

The activity of energy metabolism-related enzymes was investigated (Figure 2). The activity of CCO, H^+^-ATPase, and Ca^2+^-ATPase in soybean hypocotyl cultures under UV-B radiation significantly increased by 50.29%, 67.31%, and 25.53%, respectively. The activity of SDH and H^+^-ATPase in cotyledon suspension cultures significantly increased by 29.23% and 17.21%, respectively.

### 3.3. The Accumulation of Secondary Metabolites in Cell Suspension Cultures

As shown in Figure 3a,b, free phenolics and flavonoids accumulated in cell suspension cultures under UV-B radiation. Bound phenolics and flavonoids only accumulated in cotyledon suspension cultures. The total phenolic contents in soybean hypocotyl and cotyledon suspension cultures significantly increased by 21.18% and 91.29% compared with the control, respectively. Also, the total flavonoid contents in soybean hypocotyl and cotyledon suspension cultures increased by 13.92% and 22.82%, indicating that cotyledon suspension cultures displayed a higher sensitivity to UV-B radiation.

As shown in Figure 3c,d, the antioxidant capacity of free phenolics was greatly affected by UV-B radiation. In cotyledon suspension cultures, the ABTS and DPPH free radical scavenging ability significantly improved by 41.28% and 49.02%, respectively. The DPPH free radical scavenging ability in hypocotyl suspension cultures also improved by 33.84% under UV-B radiation.

### 3.4. Isoflavone Composition and Quantification of Cell Suspension Cultures

As shown in Table 3, the total isoflavone contents in hypocotyl and cotyledon suspension cultures significantly increased by 19.80% and 91.21% under UV-B radiation, respectively. Isoflavones mainly existed in the form of glycosides and malonyl glycosides (Table 3 and Appendix A). Five isoflavones were detected in hypocotyl suspension cultures, and four major isoflavones were found in cotyledon suspension cultures. In addition, daidzin, malonyl glycitin, and genistein were only found in cotyledon suspension cultures (UV-B group). Interestingly, UV-B radiation induced the degradation of malonyl genistein in cell suspension cultures.

### 3.5. The Key Enzyme Activities and Relative Expression of Isoflavone Synthesis

The effects of UV-B on the activity of key enzymes involved in isoflavone biosynthesis in cell suspension cultures are shown in Figure 4A–G. In hypocotyl suspension cultures, the activity of PAL, C4H, and 4CL reached their maxima at 4 h after UV-B radiation, with significant increases of 31.34%, 30.72%, and 20.69%, respectively. The activity of CHS, CHR, CHI, and IFS significantly increased by 52.70%, 59.49%, 27.61%, and 33.49% at 0 h after UV-B radiation, respectively. In soybean cotyledon suspension cultures, PAL activity significantly increased by 267.88% at 8 h after UV-B radiation. The activity of C4H and 4CL reached their maxima at 4 h after UV-B radiation, displaying significant increases of 22.47% and 13.18%. CHS, CHR, CHI, and IFS activity decreased at 8 h after UV-B radiation, followed by a rapid increase at 12 h after UV-B radiation.

The genes’ expression was generally consistent with their activities; however, a time lag phenomenon was observed among them (Figure 4). The key genes showed significant changes at 0–8 h after UV-B radiation and had no significant difference after 12 h in soybean suspension cultures compared with the control. Among them, *PAL* expression significantly increased at 0 and 8 h after UV-B radiation. The relative expression of *C4H* and *4CL* was higher at 4–8 h after UV-B radiation. In addition, the relative expression of *CHS*, *CHR2*, *CHR4*, *CHI4A*, and *IFS1* was regulated by UV-B in hypocotyl suspension cultures, while *CHR4* and *CHI1A* expression significantly increased in cotyledon suspension cultures compared with the control.

## 4. Discussion

Soybean suspension cells were composed of a large number of cell clusters and single cells, with high cellular vitality (Figure 1a–d). ATP and AMP were identified as the main energy reserves in cell suspension cultures (Table 2). The stability of cell membranes and the physiological metabolism of soybean suspension cells were affected by UV-B radiation (Figure 1e,f). SDH, CCO, and ATPases were activated by UV-B radiation, promoting ADP production and phosphorylation (Figure 2 and Table 2). The energy metabolism of cell suspension cultures plays a crucial role in cell proliferation and material accumulation under abiotic stress [21]. The production and consumption of energy are regulated by key enzymes involved in oxidative phosphorylation and ATP production [22]. As the functional protein in cell membranes, H^+^-ATPase promotes the hydrolysis of ATP into ADP with the release of energy. The balance and transport of intracellular Ca^2+^ are regulated by Ca^2+^-ATPase to ensure energy production [23]. SDH and CCO, as key enzymes in the TCA cycle and mitochondrial respiratory chain, affect cellular energy production and transduction [24]. The UV-B signal captured by soybean cells regulated the activity of ATPases, SDH, and CCO (Figure 2), thereby promoting the production and phosphorylation of ADP to maintain intracellular ATP and energy charge levels. The high energy metabolism level of cell suspension cultures not only ensured normal cell proliferation, but also sustained the energy required for secondary metabolite synthesis, like isoflavones. Hence, UV-B radiation promoted the production and phosphorylation of ADP in cell cultures by regulating the activity of ATPases, SDH, and CCO, thereby providing energy for cell growth and material accumulation.

Phenolics and flavonoids accumulated in cell suspension cultures under UV-B radiation (Figure 3). Interestingly, the suspension cultures of cotyledon were more sensitive to UV-B. As shown in Table 3, the isoflavone contents showed significant increases by 19.80% and 91.21% in hypocotyl and cotyledon suspension cultures treated with UV-B, respectively. The content of isoflavones could be as high as 700 µg/g DW. Compared to germinated soybeans, the yield of isoflavones was higher in cell suspension cultures [5]. Glycosidic and malonyl glycosidic isoflavones were the main isoflavone components. Soybean cell suspension cultures were undifferentiated thin-walled cells, and their gene expression was more susceptible to UV-B radiation compared with the entire plant (Figure 4). Glycosidic isoflavones are commonly used to form thin-walled layers or organelle membranes of cells to maintain structural stability [25]. Wang et al. [15] observed that malonyl glycosidic isoflavones accumulated in soybean calluses treated with UV-B. Liu et al. [26] also found that the glycosylation of isoflavones was effected by UV-B radiation in two Astragalus plants. Furthermore, UV-B induced the synthesis of glycoside flavonols in peach fruit [27]. These facts imply that UV-B stimulates the synthesis of isoflavones in cell suspension cultures, particularly bound isoflavones.

The biosynthesis of individual isoflavones was regulated by UV-B radiation with tissue specificity (Table 3). UV-B radiation stimulated the synthesis of malonyl daidzin, malonyl glycitin, and daidzin in cell suspension cultures. Interestingly, the synthesis of genistin mainly occurred in soybean hypocotyl cells, while daidzein, genistein, and glycitin were mainly present in cotyledon cells. UV-B radiation induces the tissue-specific accumulation of secondary metabolites in plant cells [28]. As the nutrient storage organ of the soybean plant, the cotyledon contains abundant macromolecular nutrients such as starch and protein [29]. UV-B radiation activates the expression of various endogenous enzymes in plant cells, which leads to the degradation of macromolecular nutrients into small molecules, including amino acids and glycosides [30], providing energy and essential nutrients for cells. Thus, there were plentiful isoflavones in soybean cotyledon suspension cultures, including aglycone isoflavone (daidzein and genistein). In contrast, the hypocotyl is the initial stem that emerges from the cotyledon, which transmits energy and material for the plant [29]. The formation process of the soybean hypocotyl involves tissue fibrosis [19], leading to a higher content of bound isoflavones. These compounds are often used to form cell walls and organelle membranes to protect plant cells. UV-B radiation disrupts the stability of cell membranes, resulting in the production of glycosidic and malonyl glycosidic isoflavones specifically in soybean hypocotyl cells [11]. Therefore, UV-B radiation promoted the synthesis of individual isoflavones in suspension cultures induced by different organs of soybean.

The enzymes of isoflavone biosynthesis were activated by UV-B radiation (Figure 4). The activity of six enzymes was higher at 4–8 h after UV-B radiation, and their relative expression was upregulated after UV-B radiation for 2 h. There was a temporal lag between gene expression and the activity of key enzymes in response to UV-B, indicating that six key genes of isoflavone biosynthesis showed a transient UV-B dose-dependent activation, and their expression was an early response. A delay was also detected in mRNA, protein, and isoflavone accumulation in Arabidopsis seedlings [31]. As the initial enzymes of isoflavonoid synthesis, PAL, C4H, and 4CL regulate the early stages of isoflavone biosynthesis, providing precursor substances like 4-coumaryl CoA [19]. Newton et al. [32] used CRISPR-guided DNA methylation to downregulate *PAL*, with the result of a decrease in total flavonoids in Taxus baccata L. cell cultures. Bamneshin et al. [33] found that light-induced activation of *PAL* transcription increased the content of total phenolics. Additionally, the activated CHS, CHR, CHI, and IFS further catalyzed substrates to promote the synthesis of isoflavone monomers [19]. Also, the upregulated expression of these genes promoted isoflavone synthesis in cell cultures, including Sophora flavescens [34], Dalbergia odorifera [35], and other plants. In sum, UV-B radiation induced isoflavone accumulation in cell cultures by regulating key enzymes involved in isoflavone biosynthesis at the transcriptional and substrate–enzyme levels.

## 5. Conclusions

To sum up, UV-B radiation modulated the growth and energy metabolism of soybean suspension cells, and promoted the accumulation of phenolic compounds in soybean cell suspension cultures. UV-B radiation upregulated the mRNA expression levels and enzyme activities of seven key enzymes, including PAL, C4H, and CHS. This upregulation ultimately induced the biosynthesis of isoflavones in soybean cell suspension cultures. Interestingly, the cotyledon suspension cultures were more sensitive to UV-B radiation. Notably, there were significant differences in the composition of isoflavones in soybean suspension cultures derived from different organs. Glycosidic and malonyl glycosidic isoflavone contents were higher in the soybean hypocotyl suspension cultures, whereas aglycone isoflavone was exclusively synthesized in the soybean cotyledon suspension cultures.

## Figures and Tables

**Figure 1 foods-13-02385-f001:**
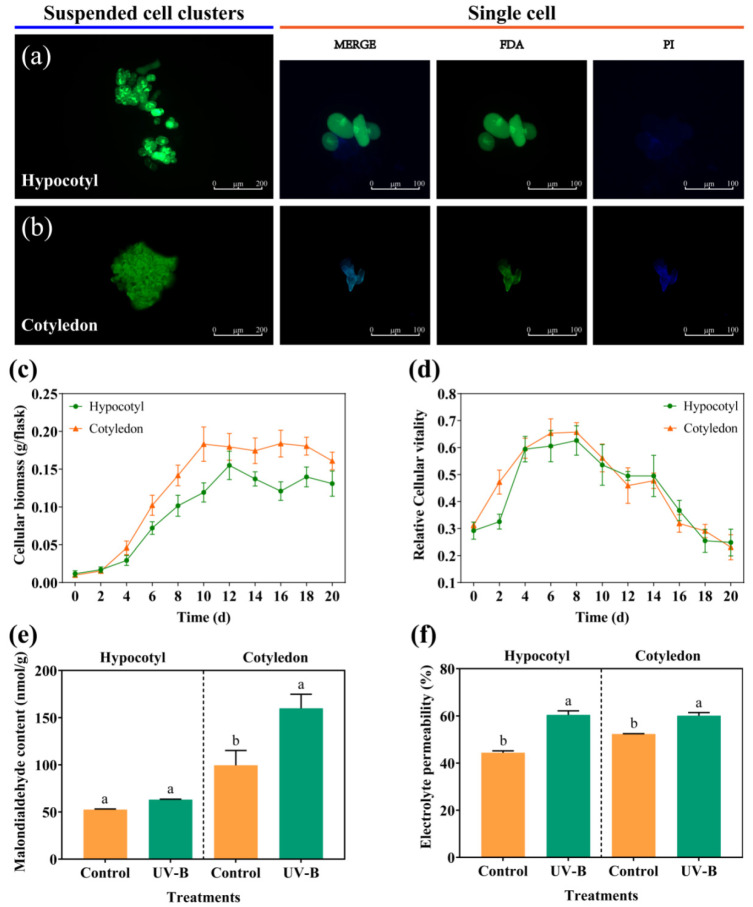
Cell morphology and physiological metabolism in cell suspension cultures. (**a**,**b**): cell morphology; (**c**): cellular biomass; (**d**): cellular vitality; (**e**): MDA content; (**f**): electrolyte permeability. UV-B: the cultures were exposed to 40 μW/cm^2^ UV-B for 2 h, followed by a 12 h incubation in the dark. Control: the cells were cultured for 14 h in the dark. The lowercase letters indicate a significant difference at *p* < 0.05 between UV-B and the control; *t*-test was used for data analysis.

**Figure 2 foods-13-02385-f002:**
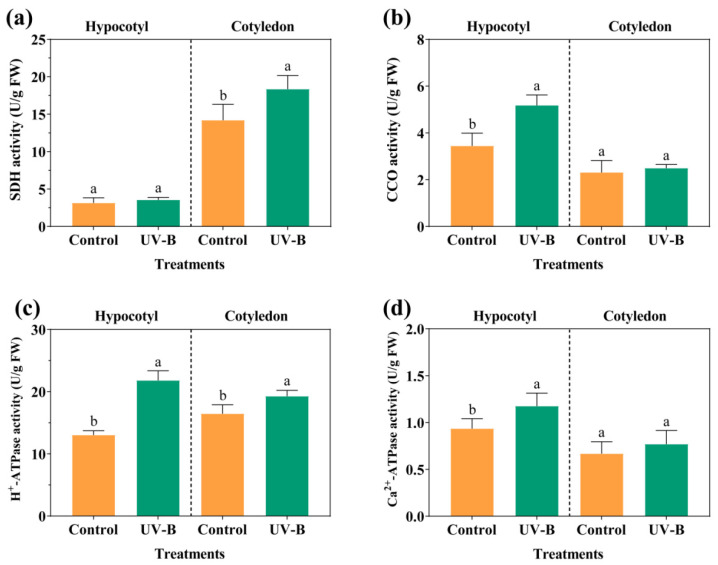
Effects of UV-B on SDH (**a**), CCO (**b**), H^+^-ATPase (**c**), and Ca^2+^-ATPase (**d**) activity in cell suspension cultures. The lowercase letters indicate a significant difference at *p* < 0.05 between treatments, and the *t*-test was used for data analysis.

**Figure 3 foods-13-02385-f003:**
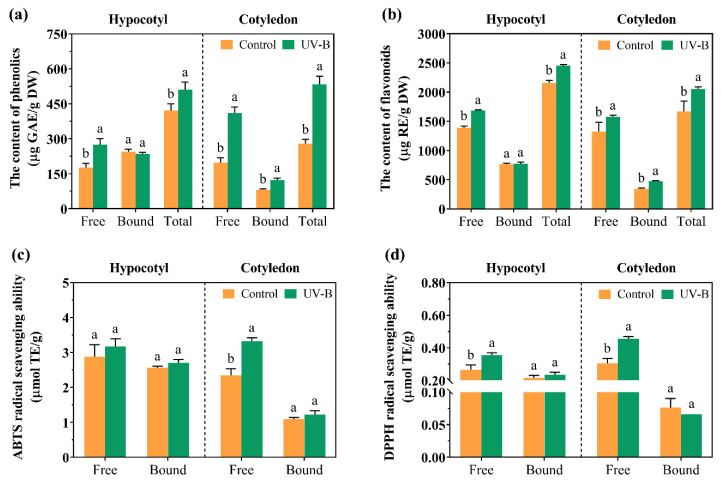
Effects of UV-B on the phenolic (**a**) and flavonoid contents (**b**) and ABTS (**c**) and DPPH (**d**) radical scavenging activity in cell suspension cultures. The lowercase letters indicate a significant difference at *p* < 0.05 between treatments, and the *t*-test was used for data analysis.

**Figure 4 foods-13-02385-f004:**
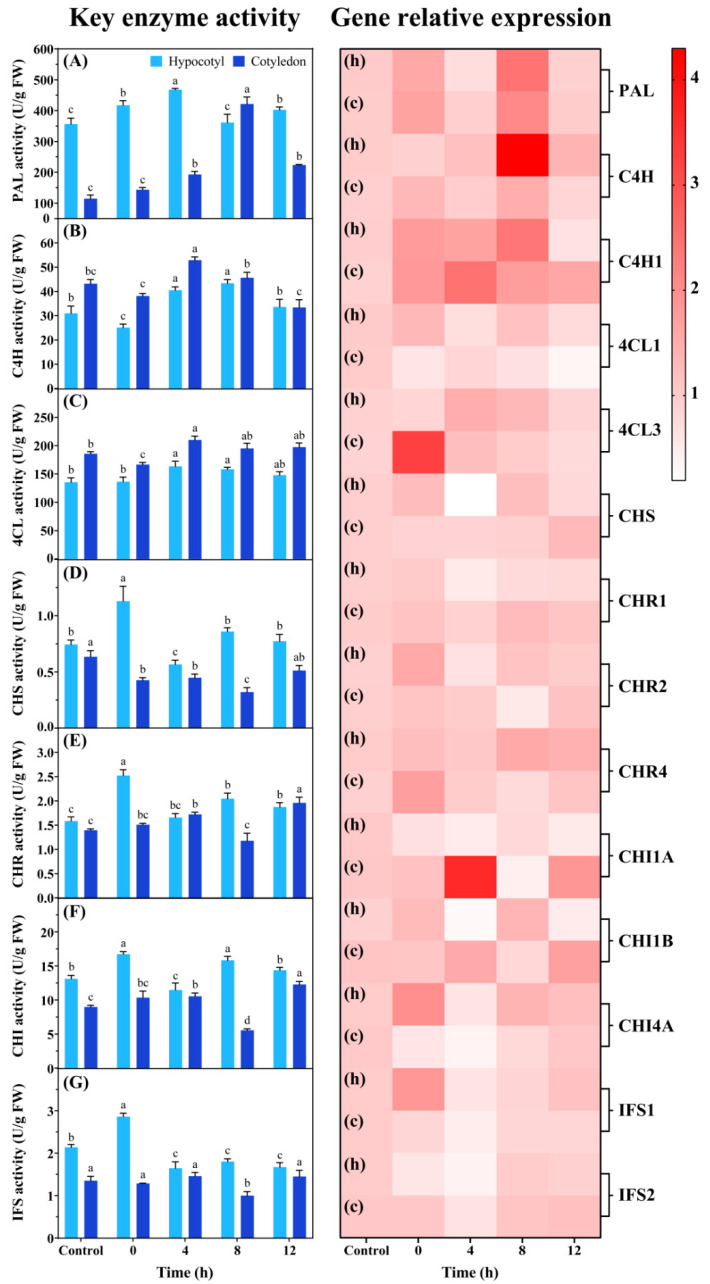
Effects of UV-B on the activity and relative expression of key enzymes in cell suspension cultures. (**A**): PAL activity; (**B**): C4H activity; (**C**): 4CL activity; (**D**): CHS activity; (**E**): CHR activity; (**F**): CHI activity; (**G**): IFS activity. h and c represent soybean suspension cultures induced by hypocotyls and cotyledons as explants, respectively. The lowercase letters indicate a significant difference at *p* < 0.05 between treatments. One-way ANOVA was used for data analysis.

**Table 1 foods-13-02385-t001:** Primers used for qRT-PCR.

Gene	Primer Name	Primer Sequences (5′→3′)	Accession Number
*PAL*	Sense	GCTAAGAAGTTGCATGAGATTGA	NM_001357058.1
	Ant-sense	TCATTGACAAGCTCAGAGAATTG	
*C4H*	Sense	CGATTTGGCCAAAAAATTCGGTG	NM_001250388.3
	Ant-sense	CCTTTCCGGTGAAGATGTCG	
*C4H1*	Sense	CGTAGAATTTGGCTCTCGCCC	NM_001371219.1
	Ant-sense	GCTGAAGCCGCCTTCTGAT	
*4CL1*	Sense	CGGTGAAATTTGCATAAGAGGC	NM_001250821.2
	Ant-sense	AATCCTTTGTATTTGATCAATTCCT	
*4CL3*	Sense	ATCTCCAACCACCTCCCT	NM_001250341.2
	Ant-sense	GGATTCCGAGGTTGGACAAT	
*CHS*	Sense	GCTATTGATGGACACCTTCG	NM_001371381.1
	Ant-sense	ACCAGGGTGTGCAATCCA	
*CHI1A*	Sense	CATTGGATGGTCGTGAATACGT	NM_001248290.2
	Ant-sense	TTGTAGAAAACAGTGGAGCCTG	
*CHI1B*	Sense	CATTGGATGGTCGTGAATACGT	NM_001249826.2
	Ant-sense	TTGTAGAAAACAGTGGAGCCTG	
*CHI4A*	Sense	ATCTTTGCTTGGCCATGGAAT	NM_001249853.2
	Ant-sense	AGCTGCCAACCTATCCCT	
*CHR1*	*Sense*	*ACTTCCAAGCTTTGGGTCAC*	*NM_001249044.2*
	*Ant-sense*	*AGGCCAAGTTTCTGGCACT*	
*CHR2*	*Sense*	*GATTCATTGGCCAGTGAGGC*	*NM_001367003.1*
	*Ant-sense*	*TCCACCTGATTGACAGCAGGA*	
*CHR4*	*Sense*	*AGCAGGCTCTTGGAGAAG*	*NM_001367005.1*
	Ant-sense	CCGAAGCGAATTTTGTAGAGCA	
*IFS1*	Sense	GAGAGCTGGCCTCACAGTTC	NM_001249093.2
	Ant-sense	TGCGATGGCAAGACACTACT	
*IFS2*	Sense	TGGAAGTTCGTGAGGAAG	NM_001251586.2
	Ant-sense	ATGGAGATGGTGCTGTTG	
*EF1b*	Sense	CCACTGCTGAAGAAGATGATGATG	NM_001249608.2
	Ant-sense	AAGGACAGAAGACTTGCCACTC	

**Table 2 foods-13-02385-t002:** The ATP, ADP, and AMP contents and energy charge levels in cell suspension cultures under UV-B radiation.

Part	Treatment	Content (µg/g FW)			Energy Charge
		ATP	ADP	AMP	
Hypocotyl	Control	30.59 ± 1.49 ^a^	ND	20.67 ± 0.43 ^b^	0.60 ± 0.02 ^a^
	UV-B	30.45 ± 0.57 ^a^	11.12 ± 0.37 ^a^	25.77 ± 1.11 ^a^	0.54 ± 0.01 ^b^
Cotyledon	Control	26.51 ± 1.41 ^b^	ND	32.40 ± 1.87 ^a^	0.45 ± 0.01 ^b^
	UV-B	34.18 ± 1.07 ^a^	9.47 ± 1.16 ^a^	26.96 ± 1.63 ^b^	0.55 ± 0.01 ^a^

ND means not detected. The lowercase letters in the same column indicate a significant difference at *p* < 0.05 between UV-B and the control in the same soybean suspension culture. A *t*-test was used. The data are presented as mean ± SD, *n* = 3.

**Table 3 foods-13-02385-t003:** The contents of individual isoflavones (µg/g DW) in cell suspension cultures under UV-B radiation.

Isoflavone	Hypocotyl		Cotyledon	
	Control	UV-B	Control	UV-B
Daidzin	45.19 ± 2.05 ^b^	173.33 ± 12.66 ^a^	ND	74.22 ± 2.86 ^a^
Glycitin	ND	ND	25.78 ± 1.07 ^b^	36.59 ± 0.59 ^a^
Genistin	89.66 ± 1.49 ^b^	139.57 ± 8.36 ^a^	ND	ND
Malonyl daidzin	106.54 ± 8.64 ^b^	217.20 ± 5.43 ^a^	161.47 ± 12.17 ^b^	279.28 ± 10.90 ^a^
Malonyl glycitin	54.78 ± 4.92 ^b^	98.00 ± 9.08 ^a^	ND	34.94 ± 1.69 ^a^
Malonyl genistin	297.49 ± 9.48 ^a^	83.11 ± 5.99 ^b^	83.74 ± 0.62 ^a^	68.93 ± 6.13 ^b^
Daidzein	ND	ND	27.98 ± 2.07 ^b^	47.15 ± 2.54 ^a^
Glycitein	ND	ND	ND	ND
Genistein	ND	ND	ND	30.56 ± 0.65 ^a^
Total	593.65 ± 14.42 ^b^	711.21 ± 16.08 ^a^	298.97 ± 12.51 ^b^	571.65 ± 13.10 ^a^

Effects of UV-B on the contents of individual isoflavones in soybean hypocotyl and cotyledon suspension cultures. ND means not detected. The lowercase letters in the same row indicate significant differences at *p* < 0.05 between UV-B and the control in the same callus. A *t*-test was used. The data are presented as mean ± SD, *n* = 3.

## Data Availability

The original contributions presented in this study are included in the article/Appendix A; further inquiries can be directed to the corresponding author.

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
