# Peer review of "UV-B Radiation Exhibited Tissue-Specific Regulation of Isoflavone Biosynthesis in Soybean Cell Suspension Cultures"

_foods, 2024, doi:10.3390/foods13152385_

Round 1

Reviewer 1 Report

Comments and Suggestions for Authors

Dear authors,

In general the manuscript is well structured.

However, I have several important remarks:

2. Materials and Methods

 2.1. Chemicals:

Please describe precisely, all the chemicals involved in the study. Include not only the brand, but also the purity/the analytical grade. Include the chemical you have used for the HPLC method.

2.7. Identification and quantification of isoflavonoids:

You must include the chromatographic conditions not only the citation.

"Typical HPLC chromatograms of isoflavone monomers were  shown in Figure S1." Where is this figure?  The manuscript does not contain a chromatogram. A chromatogram of the results must be included in the results section (not a chromatogram of standard solutions, but of the results).

5. Conclusions:

Is it possible to make the Conclusions more detailed?

Author Response

Dear Editor and Reviewers,

We are very grateful to you for the comments and advices on our initial manuscript (Manuscript ID is 3116472, entitled “UV-B radiation exhibited tissue-specific regulation on the isoflavone biosynthesis in soybean cell suspension cultures”). Based on the comments and advises, we have revised our manuscript carefully, and resubmitted to you. We are willing to do any further revision if necessary.

Reviewer(s)' Comments to Author:

In general the manuscript is well structured. However, I have several important remarks:

1. Chemicals: Please describe precisely, all the chemicals involved in the study. Include not only the brand, but also the purity/the analytical grade. Include the chemical you have used for the HPLC method.

Response: Thanks! We have checked and revised carefully (Lines 83-91).

2. Identification and quantification of isoflavonoids: You must include the chromatographic conditions not only the citation.

Response: Thanks! The chromatographic conditions have been included in the methodology (Lines 140-146).

3. "Typical HPLC chromatograms of isoflavone monomers were  shown in Figure S1." Where is this figure?  The manuscript does not contain a chromatogram. A chromatogram of the results must be included in the results section (not a chromatogram of standard solutions, but of the results).

Response: Thanks! The chromatograms of the standard solutions and samples were shown in the Supplementary Materials (Lines 350-355, Figure S1). The chromatograms of the 9 individual isoflavone standards were shown in Figure S1A. Figures S1B and S1C showed the chromatograms of the isoflavones present in the hypocotyl and cotyledon cell cultures, respectively. The isoflavone compounds in the cell culture extracts were further analyzed and identified based on these chromatograms.

4. Conclusions:Is it possible to make the Conclusions more detailed?

Response: Thanks! We have revised and perfected our manuscript carefully (Lines 339-349).

Reviewer 2 Report

Comments and Suggestions for Authors

While presented paper is scientifically sound, it has one important issue. The same authors already published very similar paper (paper number 15 in this manuscript). It is not clear what is the novelty of this paper compared to the previous one. It seems that similar materials and similar approaches were used in both papers. 

Have the authors tested several different conditions for UV-B exposure, besides the one presented here? 

Some parts in Introduction should be in Discussion section, specifically the ones comparing the yields of isoflavones between studies.  

Comments on the Quality of English Language

The authors need to check entire manuscript since in many sentences there are hyphenated words on the same lines. Please correct this.  

Author Response

Dear Editor and Reviewers,

We are very grateful to you for the comments and advices on our initial manuscript (Manuscript ID is 3116472, entitled “UV-B radiation exhibited tissue-specific regulation on the isoflavone biosynthesis in soybean cell suspension cultures”). Based on the comments and advises, we have revised our manuscript carefully, and resubmitted to you. We are willing to do any further revision if necessary.

Reviewer(s)' Comments to Author:

1. While presented paper is scientifically sound, it has one important issue. The same authors already published very similar paper (paper number 15 in this manuscript). It is not clear what is the novelty of this paper compared to the previous one. It seems that similar materials and similar approaches were used in both papers.

Response: Thanks! The research objectives of the previous study (paper number 15) and the current manuscript are distinctly different. The aim of our previous study (paper number 15 in this manuscript) was to establish the soybean callus tissue formation system, and assess the synthetic metabolism of isoflavones in these soybean callus tissues under UV-B radiation. In contrast, the aim of the current manuscript is to explore the tissue-specific regulation on the physiological metabolism and isoflavone biosynthesis in soybean cell suspension cultures under UV-B radiation. The previous study laid the experimental foundation for this work, as suspension cells are induced from callus tissue. However, suspension cells have significantly different morphology, material and energy transfer mechanisms compared to callus tissue. Whether suspension cells respond to UV-B radiation, and how they respond, requires further study.

This current study builds upon the experimental conditions of the previous research to further explore the specific regulation of UV-B radiation on the proliferation, energy metabolism level and isoflavone biosynthesis of soybean suspension cells. Additionally, unlike callus tissue, suspension cells can achieve large-scale continuous cultivation, which is necessary for the future development of food-related applications. This study can provide a new production method for the accumulation of functional compounds such as isoflavones, potentially reducing resource waste and enabling continuous secondary metabolite production.

2. Have the authors tested several different conditions for UV-B exposure, besides the one presented here?

Response: Thanks! The intensity of UV-B radiation and the incubation time were determined through experiments as detailed in the previous paper (paper number 15 in this manuscript). The previous study investigated the effects of different UV-B radiation intensities (20, 40, and 80 μW/cm2), secondary incubation times, and plant hormones on the growth metabolism and isoflavone accumulation in soybean calluses, and optimized the culture conditions. The results showed that soybean calluses radiated with 40 μW/cm2 UV-B for 2 hours, followed by a 12-hour re-incubation in the dark, exhibited a high isoflavone content. Building upon these previous findings, the current study aims to further analyze the tissue-specific regulation of isoflavone biosynthesis in soybean cell suspensions under UV-B radiation. Therefore, we selects specific UV-B radiation intensity to treat the soybean cell suspension cultures in this manuscript. Of course, future research can also investigate investigate the effects of different soybean varieties and UV-B radiation modes on the accumulation of secondary metabolites in suspended cells.

3. Some parts in Introduction should be in Discussion section, specifically the ones comparing the yields of isoflavones between studies.

Response: Thanks! The discussion section has been supplemented with a comparison of the isoflavone content between suspension cells and germinating soybeans, elucidating the significance of suspension cell-induced isoflavone production (Lines 286-290). The introduction of this manuscript has outlined the research progress on isoflavone biosynthesis in soybean suspension cells, describing the composition and content of isoflavones in suspension cells, and emphasizing the advantages of suspension cell culture to highlight the enormous potential of soybean suspension cultures in producing bioactive compounds such as isoflavones. However, the focus in the discussion section is on the tissue-specific regulation of soybean suspension cells by UV-B radiation based on the results of this study. The aim is to demonstrate that suspension cells induced from different organs exhibit varying responses to UV-B, which can be exploited to achieve targeted regulation and production of individual isoflavones. Therefore, the part in the introduction regarding the yields of isoflavones was not fully incorporated into discussion section, and instead, some parts were supplemented in the discussion. Thank you again for your suggestion.

4. The authors need to check entire manuscript since in many sentences there are hyphenated words on the same lines. Please correct this.

Response: Thanks! we have made corrections and improvements to the manuscript.

Round 2

Reviewer 1 Report

Comments and Suggestions for Authors

Accept in present form

Reviewer 2 Report

Comments and Suggestions for Authors

I appreciate the authors for addressing all of my issues.